# Assessment of Water Quality for Aquaculture in Hau River, Mekong Delta, Vietnam Using Multivariate Statistical Analysis

Fridah Gacheri Mutea, Howard Kasigwa Nelson, Hoa Van Au, Truong Giang Huynh and Ut Ngoc Vu *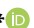

Department of Applied Hydrobiology, College of Aquaculture and Fisheries, Can Tho University, 3/2 Street, Ninh Kieu District, Can Tho City 94000, Vietnam; fgacherimutea@gmail.com (F.G.M.); hkasigwa@gmail.com (H.K.N.); avhoa@ctu.edu.vn (H.V.A.); htgiang@ctu.edu.vn (T.G.H.)
\* Correspondence: vnut@ctu.edu.vn

**Abstract:** The deterioration signs of water quality in the Hau River are apparent. The present study analyzed the surface water quality of the Hau River using multivariate statistical techniques, including principal component analysis (PCA) and Cluster Analysis (CA). Eleven water quality parameters were analyzed at 19 different sites in An Giang and Can Tho Provinces for 12 months from January to December 2019. The findings show high levels of Biological Oxygen Demand (BOD), Total Soluble Solids (TSS), and total coliform, all year round. The PCA revealed that all the water quality parameters influenced the water quality of the Hau River, hence the relevance for water sample scrutiny. The dendrogram of similarity between sampling sites showed a maximum similarity of 95.6%. The Accumulation Factor (AF) trend showed that the concentrations/values of TSS, BOD, and phosphate ($PO4^{3-}$) in the downstream were 1.29, 1.53, and 1.52 times, respectively, greater than the upstream levels. Despite most of the parameters analyzed supporting aquaculture production, caution is needed in the regulation of pollution point sources to undertake sustainable aquaculture production.

**Keywords:** pollution; river water; cluster analysis; accumulation factor; anthropogenic activities

## 1. Introduction

Rivers are one of the critical channels for human survival that have a unique role in the origin and development of human societies [1,2]. They have always been well-thought-out as valuable freshwater resources for life because most of the activities under development depend on them to date. Additionally, ancient people have thrived along with them [3]. Rivers supply water for different uses, including industry, agriculture, drinking [4], aquaculture, public water supply, and transportation, among others [3].

The surface water quality is a significant issue beyond the national boundaries, which is affected by both natural and anthropogenic sources (including factors such as atmospheric chemistry, layers below the soil surface, plants (or decomposed organic matter), and unusual agents [5]. Not forgetting, pollution sources such as the leather industry, marble factories, and fish farms [6], increasing population, industrialization, and urbanization have led to water scarcity and water quality deterioration and adversely affected both social and economic development [7]. These factors have led to contamination of the rivers, especially in developing countries, through effluents full of different chemicals from massive industrial developments, albeit meeting the needs of increasing populations [8]. Spatiotemporal analysis of surface water in the rivers is vital in protecting and conserving these aquatic resources [9]. Unfortunately, Spatiotemporal variations of water quality have been perceived to be difficult for a long time due to the nature of environmental data, which are linear; that is why statistical approaches provide reliable water quality analysis [10].

The Hau River is a branch of the Mekong Delta ecosystem, one of the richest areas of biodiversity in the world behind only the Amazon basin, therefore termed as the lifeblood of the region, which provides sustenance through supporting aquaculture and agriculture [11].

The Vietnam Mekong Delta is a pivotal contributor to the economy of Vietnam because it contributes approximately 27% of the country's Gross Domestic Product (GDP), supporting 16 million citizens (nearly 22% of Vietnam's total population), and providing about 50% of the annual rice production [12]. It is the primary agricultural production region of Vietnam due to its fertile soils and abundant water resource [13], thus contributing to export as well as the food provision for Vietnamese [14]. However, Mekong Delta is considered one of the most vulnerable catchment areas throughout the basin because of the cumulative impacts of upstream development activities. As a result of these activities, it experiences increasing salinity intrusion, which represents a critical challenge for water resources management and agriculture production despite the delta playing an essential role in the economy of Vietnam [15]. The ecosystem services of the Hau River have been overused, and the water quality is deteriorating with increasing nutrient levels and decreasing oxygen levels due to eutrophication. The Hau River faces rising pressure from overwhelming human activities such as agriculture, aquaculture, and an increased number of boats and industries [16]. In support of this, Tuan [17] pointed out that the Hau River is facing many challenges such as water quality deterioration, narrowing of natural lowlands during the urbanization processes, changes in hydrological flow characteristics, expansion of agriculture and fisheries production activities, and effects of climate change, such as the sea-level rise and further saltwater intrusion.

Seemingly to note, meaningful conclusions of water quality have been drawn from some unbiased analysis methods such as factor analysis and cluster analysis, which are critical multivariate statistical techniques [18,19]. In addition, multivariate analytical methods are also crucial in the evaluation and characterization of water quality to analyze spatiotemporal variations that have resulted from both natural and anthropogenic activities [20,21]. This is because water quality deterioration can only be solved through scientific and reasonable water quality assessment methods as a significant basis for ensuring proper water quality management and assessment [22]. The present study aims to address the issue by identifying the water parameters accountable for spatiotemporal variation of water quality in the Hau River using principal component analysis (PCA) and cluster analysis (CA). Additionally, the study aimed at assessing the water quality of the river at the 19 sampling sites every month in the year 2019 to ensure it is safe for aquaculture practice.

## 2. Materials and Methods

### 2.1. Study Location

The present study was carried out on the Hau River in An Giang and Can Tho Provinces, Vietnam, as illustrated in Figure 1. The Hau River is one of the two main branches of the Mekong River that flows to the Mekong Delta through the Cambodian border [22]. Among the two branches, the Hau River is located at the most southern side of the Mekong and runs through the Vietnamese territory in An Giang Province, An Phu District, Khanh An commune, as it flows into the South China sea [23]. It splits into two sub-branches, namely: Tran De and Dinh An as it approaches the sea [22]. A total quantity of water amounting to approximately 20 m$^3$ per year flows into the sea from the Hau River [24], thus accounting for about 41% of the total Mekong water discharge, whereby 30% of the discharge flows through the Tran De channel. In contrast, 70% of it flows through Dinh An channel [22]. This positions the Hau River as having the highest water discharge compared to other rivers in Vietnam because it can reach the extent of draining around 90% of Mekong River's peak floodwater, not forgetting its total annual flow of nearly 215 billion m$^3$ [25].

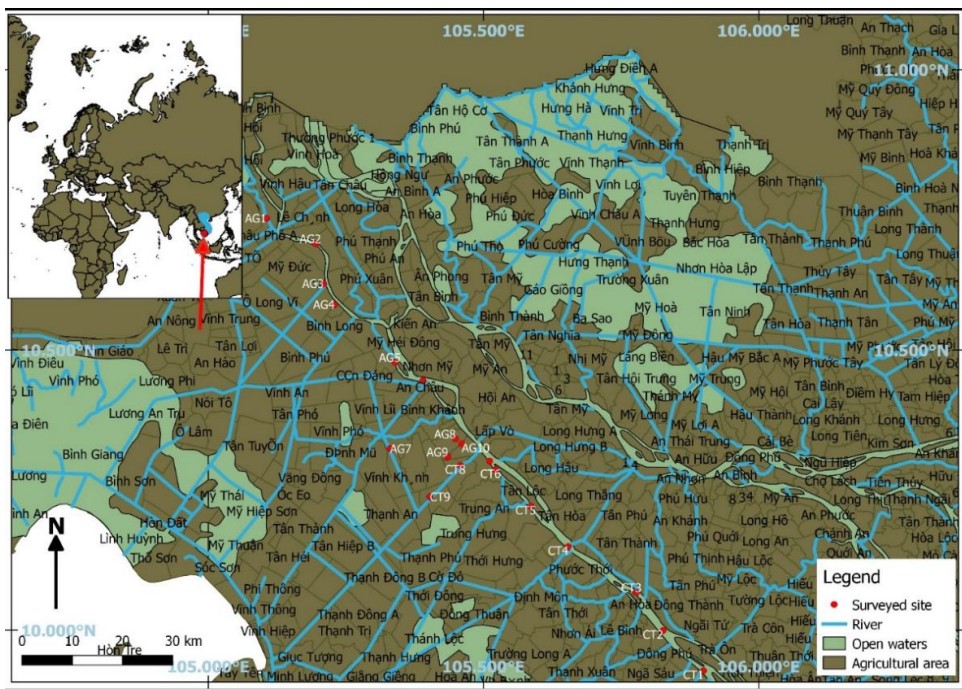

**Figure 1.** Location of the sampling sites in An Giang and Can Tho Provinces.

The Hau River has a length of 225,000 m and a width of approximately 60–300 m upon entering Vietnam, although the river widens bit by bit as it flows to the Sea. The Hau River has an average depth of 10–20 m while the maximum is over 40 m. Unfortunately, sedimentation has caused a decrease in sea depth as the river nears the sea [25]. It has a large flow velocity ranging from 1.0 to 2.98 m/s [26], and that at the mouth of the river is as high as 1.5 m/s despite the low water exchange between the river and the Delta. The Hau River serves as a source of water for very many people in An Giang and Can Tho Provinces for various purposes such as domestic, industrial, aquaculture, and agricultural uses. Therefore, the study results will help share information that will help the environmental managers in Can Tho and An Giang address the deteriorating status of water quality in the Hau River and give recommendations that will ensure its sustainable management. The description of the 19 sampling sites is as follows, with AG1 being upstream and CT1 downstream (Table 1).

### 2.2. Sampling and Analysis

A total of 19 sites were sampled from An Giang and Can Tho Provinces. The sampling was conducted from January 2019 to December 2019 for 2–3 days every month, and data were obtained by measuring all the water quality parameters in the various selected sites. The water quality parameters that were assessed consisted of temperature, pH, Dissolved oxygen, salinity, biological oxygen demand (BOD), alkalinity, total ammonia nitrate (TAN), nitrate (N-NO$_3^-$), total suspended solids (TSS), phosphate (P-PO$_4^{3-}$), and total coliform. The samples were taken proportionately to ensure that nutrient pulses were captured during the rainy season with run-off events. Notably, the samples were collected in the morning between 8 am and 12 noon from each sampling site and then placed in pre-rinsed and acid-washed 100 mL plastic bottles, which were well-labeled with the corresponding sampling station. Each bottle was filled with water sample then tightly closed using a plastic bag and immediately placed in a cool box with ice at $4^0$C to ensure proper preservation of the samples while in the field. Upon arrival at the laboratory, the samples were refrigerated to avoid external contamination until the time of the analysis on the same day of sample collection or the next day. Temperature, pH, salinity, and Dissolved oxygen were measured directly on site while the other parameters were

analyzed at the laboratory of Can Tho University, applying fundamental methods according to American Public Health Association (APHA) standard methods [27]. Temperature, pH, and Dissolved oxygen were measured using a YSI meter, while salinity was measured using a refractometer in situ.

**Table 1.** Description of the sampling sites.

| Site Name | Code | Longitude (N) | Latitude (E) | Description |
|---|---|---|---|---|
| Vĩnh Ngươn | AG1 | 10°44.103′ | 105°06.33′ | fish cage area, cultured ponds, residences |
| Cồn Khánh Hòa | AG2 | 10°41.406′ | 105°11.704′ | intensive fish culture area, mainly in the surrounding regions of the islet |
| Vịnh Tre | AG3 | 10°37.117′ | 105°12.574′ | fish ponds and residences |
| Cầu chữ S | AG4 | 10°34.875′ | 105°13.768′ | fish ponds and residences |
| Bến đò Rạch Gọc | AG5 | 10°28.706′ | 105°20.358′ | fish ponds in Binh Thuy islet |
| Bến đò Sơn Đốt | AG6 | 10°26.751′ | 105°23.408′ | fish ponds and residences |
| Kênh Ông Cò | AG7 | 10°19.438′ | 105°19.811′ | fish ponds and residences |
| Kênh Tây An | AG8 | 10°20.502′ | 105°26.956′ | fish ponds and residences |
| Kênh Cái Sao 2 | AG9 | 10°18.576′ | 105°26.122′ | fish ponds and residences |
| Kênh Cái Sao 1 | AG10 | 10°19.969′ | 105°27.644′ | fish ponds and agriculture activities |
| Thạnh Mỹ-Vĩnh Thạnh | CT9 | 10°14.276′ | 105°24.164′ | fish ponds and residences |
| Cái Sắn | CT8 | 10°17.659′ | 105°27.483′ | fish ponds and residences |
| Bến phà Bò Ót | CT7 | 10°18′07.7″ | 105°30′40.9″ | fish culture area, residences, industrial zone |
| Bến phà Trà Uối | CT6 | 10°17.201′ | 105°31.322′ | fish culture area, residences, industrial zone |
| Thuận Hưng | CT5 | 10°13.290′ | 105°35.155′ | fish ponds, fish cages, and residences |
| Thới An | CT 4 | 10°08.964′ | 105°39.236′ | fish culture area and residences |
| Cồn Khương | CT 3 | 10°04.044′ | 105°46.671′ | fish culture area and residences |
| Cái Cui | CT 2 | 09°59.564′ | 105°49.579′ | Industrial activities |
| Cái Côn | CT 1 | 09°55.653′ | 105°53.990′ | Residences |

BOD was measured with BOD sensors and incubator through inoculation and dilution using the APHA 5210B method. As for alkalinity, APHA 2320B acid titration was done using a digital micropipette, and TAN was measured using a spectrophotometer following APHA 4500 method. $N\text{-}NO_3^-$ and $P\text{-}PO_4^{3-}$ were also measured using spectrophotometer although $N\text{-}NO_3^-$ was through cadmium reduction method in APHA 4500E-$NO_3^-$ while $P\text{-}PO_4^{3-}$ was through silicon chloride reaction highlighted in APHA 4500-P. Other analyses included TSS by mass measurement using filter paper (filtration and drying at 103–105 °C) as emphasized in APHA 2540D and total coliform, which was determined by standard Most Probable Number (MPN) using an incubator and test tubes as described in APHA 9221 method. Each sample was analyzed for the following parameters: BOD, alkalinity, TAN, TSS, $N\text{-}NO_3^-$, $P\text{-}PO_4^{3-}$, and total coliform. Before calculating the standard deviation and the mean value, each sample of the measured parameter was analyzed in triplicate. Additionally, the instruments and probe meters were calibrated before analysis, and the standards and blanks were given the same treatment as the representative river water samples to ensure accuracy and minimal matrix interference during laboratory analysis.

*2.3. Data Analysis*

The water quality parameters in the study were compared with QCVN 08-MT: 2015/BTNMT National technical regulation on surface water quality [28] and other references with standard values for water quality parameters in general aquaculture practices. Importantly, the data for water quality parameters of the water samples were analyzed using descriptive statistics and presented as mean values in Microsoft Excel and MINITAB 19.2020.1.0 software applications. The degree of pollution from anthropogenic and natural factors was estimated using the Accumulation Factor (AF) which is the ratio of the average level of a given water quality parameter downstream (after source discharge) to the corresponding intermediate level of the same parameter upstream (before the source discharge) [29].

Additionally, the degree of river recovery capacity (RRC) for the Hau River was calculated using the mathematical equation given by Fakayode [21], which is as follows:

$$\text{RRC} = \frac{(S0 - S1)}{S0} \times 100 \text{ (expressed in \%)}$$

where $S0$ is the concentration of a parameter downstream (i.e., immediately after the discharge point), and $S1$ is the conforming average level in the upstream where the water is relatively unpolluted.

### 2.4. Multivariate Statistical Methods

Data were analyzed using a one-way analysis of variance (ANOVA) at a 0.05% level of significance to meet the objective of evaluating the significant differences among the 19 sites for all the water quality parameters. Cluster analysis and principal component analysis were the two multivariate techniques used to explore groups and sets of variables with similar properties. These allowed us to simplify the description of observations made by finding the patterns and structure in the case of complex or chaotic data [30]. These multivariate techniques helped facilitate the consistent evaluation of the multiple variables examined during the different periods and sampling points. All the statistical analyses were performed using MINITAB 19.2020.1.0 software application.

### 2.5. Principal Component Analysis (PCA)

The principal component analysis is a technique that reduces data and suggests how many variables are essential to explain the observed variation in the data. This is because PCA can be used to reduce the number of variables and, at the same time, indicate the same amount of variation with fewer variables, also known as principal components [31]. In classical PCA, the PC has a more exceptional contribution to explaining the variation observed in the original data, meaning that the eigenvalue is more substantial. Besides, PCA is instrumental in explaining any correlation between the observations made regarding the fundamental factors that are not directly observable [32]. All the concentrations of the parameters were log-transformed before modeling to have a closer distribution, and statistical conclusions were made based on a multiparametric model. We have used ANOVA, PCA, and CA to assess the impact of natural factors and anthropogenic activities and Spatiotemporal variations on the physicochemical characteristics of the Hau River.

### 2.6. Cluster Analysis (CA)

Cluster analysis is a multivariate statistical technique that helps assemble objects based on their similarity by grouping survey locations based on the criterion of surface water quality. The water quality parameters and sampling sites were grouped based on similarities and dissimilarities of water quality using Ward's method using Euclidean distance, representing the difference between the environmental samples and analytical values [33]. The results from the cluster analysis were then presented in a dendrogram [34]. The dendrogram illustrates the cluster processes in a visual summary, showing a picture of the groups and their closeness with a dramatic reduction in the dimensionality of the original data [35].

## 3. Results and Discussion

### 3.1. Summary of Water Quality Parameters in An Giang and Can Tho Provinces

Water quality parameters are vital elements in aquaculture, and therefore they should be monitored to ensure they are within the desired range. The mean values of physicochemical parameters at different sampling sites in the Hau River during the period of 12 months (January 2019–December 2019) are presented in Table 2.

**Table 2.** Water quality of the Hau River in 2019 in An Giang (G1–AG10) and Can Tho Provinces (CT1–CT9).

| Parameter | AG1 | AG2 | AG3 | AG4 | AG5 | AG6 | AG7 | AG8 | AG9 | AG10 | Desirable Range [28] |
|---|---|---|---|---|---|---|---|---|---|---|---|
| Temperature (°C) | 30.8 ± 1.23 $^a$ | 30.6 ± 1.48 $^a$ | 30.5 ± 1.31 $^a$ | 30.4 ± 1.24 $^a$ | 30.3 ± 1.38 $^a$ | 30.5 ± 1.35 $^a$ | 31.0 ± 1.27 $^a$ | 30.7 ± 1.34 $^a$ | 30.7 ± 1.21 $^a$ | 30.5 ± 1.35 $^a$ | 20–32 |
| pH | 7.2 ± 0.21 $^{bc}$ | 7.4 ± 0.21 $^{ab}$ | 7.3 ± 0.26 $^{abc}$ | 7.4 ± 0.18 $^{ab}$ | 7.6 ± 0.20 $^a$ | 7.3 ± 0.26 $^{abc}$ | 7.0 ± 0.13 $^c$ | 7.0 ± 0.23 $^c$ | 7.0 ± 0.15 $^c$ | 7.4 ± 0.18 $^{ab}$ | 6.5–9 |
| Alkalinity (mg/L) | 60.7 ± 14.39 $^a$ | 63.3 ± 16.89 $^a$ | 63.2 ± 17.99 $^a$ | 64.1 ± 16.76 $^a$ | 63.8 ± 15.48 $^a$ | 63.4 ± 15.79 $^a$ | 64.5 ± 16.3 $^a$ | 67.5 ± 18.72 $^a$ | 66.1 ± 16.39 $^a$ | 63.4 ± 15.38 $^a$ | 25–100 |
| DISSOLVED OXYGEN (mg/L) | 4.8 ± 0.60 $^{bcde}$ | 5.6 ± 0.65 $^{ab}$ | 5.7 ± 0.73 $^a$ | 5.4 ± 0.57 $^{abcd}$ | 5.7 ± 0.39 $^a$ | 5.5 ± 0.62 $^{abc}$ | 4.0 ± 0.61 $^{ef}$ | 4.9 ± 0.86 $^{abcde}$ | 4.5 ± 0.74 $^{def}$ | 5.4 ± 0.63 $^{abcd}$ | 5–15 |
| Sal (mg/L) | 141.5 ± 27.32 $^{abcde}$ | 138.7 ± 36.1 $^{abcde}$ | 148.4 ± 32.81 $^{abc}$ | 141.3 ± 26.73 $^{abcde}$ | 144.6 ± 32.5 $^{abcd}$ | 127.5 ± 25.63 $^{cdef}$ | 147.7 ± 26.77 $^{abc}$ | 136.8 ± 21.89 $^{abcdef}$ | 152.1 ± 41.3 $^{ab}$ | 153.6 ± 41 $^a$ | 0–5000 |
| TSS (mg/L) | 37.9 ± 14.56 $^{ab}$ | 33.4 ± 24.15 $^{ab}$ | 45.0 ± 21.1 $^{ab}$ | 35.4 ± 12.58 $^{ab}$ | 30.1 ± 14.05 $^{ab}$ | 32.1 ± 22.92 $^{ab}$ | 48.5 ± 24.53 $^{ab}$ | 40.4 ± 21.16 $^{ab}$ | 41.3 ± 23.26 $^{ab}$ | 27.6 ± 10.71 $^b$ | 25–150 |
| TAN (mg/L) | 0.2 ± 0.10 $^{ab}$ | 0.1 ± 0.05 $^{ab}$ | 0.2 ± 0.07 $^{ab}$ | 0.2 ± 0.16 $^{ab}$ | 0.2 ± 0.09 $^{ab}$ | 0.2 ± 0.19 $^{ab}$ | 0.4 ± 0.23 $^a$ | 0.3 ± 0.17 $^{ab}$ | 0.2 ± 0.14 $^{ab}$ | 0.2 ± 0.16 $^{ab}$ | <0.01 |
| BOD (mg/L) | 3 ± 0.56 $^{cde}$ | 3.0 ± 0.75 $^{cde}$ | 3.0 ± 0.54 $^{cde}$ | 2.7 ± 0.67 $^e$ | 2.9 ± 0.86 $^{de}$ | 2.9 ± 0.92 $^{cde}$ | 2.9 ± 0.75 $^{cde}$ | 2.7 ± 0.74 $^e$ | 3.0 ± 0.98 $^{cde}$ | 3.0 ± 1.11 $^{cde}$ | 1–2 |
| NO$_3^-$ (mg/L) | 0.2 ± 0.10 $^a$ | 0.2 ± 0.06 $^a$ | 0.2 ± 0.10 $^a$ | 0.2 ± 0.10 $^a$ | 0.2 ± 0.08 $^a$ | 0.2 ± 0.09 $^a$ | 0.2 ± 0.08 $^a$ | 0.2 ± 0.12 $^a$ | 0.2 ± 0.06 $^a$ | 0.2 ± 0.08 $^a$ | 0.1–4.5 |
| PO$_4^{3-}$ (mg/L) | 0.2 ± 0.16 $^a$ | 0.1 ± 0.12 $^a$ | 0.2 ± 0.14 $^a$ | 0.2 ± 0.18 $^a$ | 0.1 ± 0.05 $^a$ | 0.1 ± 0.17 $^a$ | 0.1 ± 0.06 $^a$ | 0.2 ± 0.14 $^a$ | 0.2 ± 0.08 $^a$ | 0.2 ± 0.19 $^a$ | 0.05–0.5 |
| TC (MPNx $10^3$/100 mL) | 24.8 ± 25.66 $^a$ | 14.2 ± 13.26 $^a$ | 21.1 ± 33.96 $^a$ | 31.5 ± 36.7 $^a$ | 30.1 ± 39.2 $^a$ | 39.6 ± 53 $^a$ | 17.9 ± 30.2 $^a$ | 20.1 ± 15.64 $^a$ | 24.2 ± 23.62 $^a$ | 19.9 ± 17.47 $^a$ | 0.1–5 |

| Parameter | CT1 | CT2 | CT3 | CT4 | CT5 | CT6 | CT7 | CT8 | CT9 | | |
|---|---|---|---|---|---|---|---|---|---|---|---|
| Temperature (°C) | 29.9 ± 1.94 $^a$ | 30.1 ± 1.22 $^a$ | 30.2 ± 1.25 $^a$ | 30.3 ± 1.15 $^a$ | 30.5 ± 1.16 $^a$ | 30.4 ± 1.25 $^a$ | 30.5 ± 1.35 $^a$ | 30.5 ± 1.06 $^a$ | 30.5 ± 1.08 $^a$ | | 20–32 |
| pH | 7.4 ± 0.36 $^{ab}$ | 7.4 ± 0.27 $^{ab}$ | 7.2 ± 0.24 $^{bc}$ | 7.2 ± 0.21 $^{bc}$ | 7.2 ± 0.22 $^{bc}$ | 7.3 ± 0.16 $^{abc}$ | 7.3 ± 0.15 $^{abc}$ | 7.2 ± 0.29 $^{bc}$ | 7.2 ± 0.29 $^{bc}$ | | 6.5–9 |
| Alkalinity (mg/L) | 63.4 ± 15.74 $^a$ | 66.3 ± 12.98 $^a$ | 64.0 ± 14.43 $^a$ | 65.4 ± 14.38 $^a$ | 63.6 ± 14.28 $^a$ | 62.4 ± 16.85 $^a$ | 64.9 ± 12.66 $^a$ | 63.4 ± 15.37 $^a$ | 63.4 ± 15.37 $^a$ | | 25–100 |
| DISSOLVED OXYGEN (mg/L) | **4.2 ± 0.72 $^{ef}$** | **4.6 ± 0.93 $^{def}$** | **4.6 ± 0.53 $^{cde}$** | **4.7 ± 0.57 $^{cde}$** | **4.2 ± 0.77 $^{ef}$** | **4.6 ± 0.44 $^{cde}$** | **4.6 ± 0.39 $^{cde}$** | **4.6 ± 0.52 $^{de}$** | **3.7 ± 0.34 $^f$** | | 5–15 |
| Sal (mg/L) | 131.0 ± 26.14 $^{abcdef}$ | 122.8 ± 30.55 $^{def}$ | 113.6 ± 28.72 $^f$ | 126.8 ± 37 $^{cdef}$ | 128.2 ± 23.82 $^{bcdef}$ | 124.3 ± 28.77 $^{cdef}$ | 127.6 ± 21.91 $^{bcdef}$ | 117.3 ± 32.41 $^{ef}$ | 129.8 ± 27.53 $^{abcdef}$ | | 0–5000 |
| TSS (mg/L) | 45.9 ± 31.06 $^{ab}$ | 46.3 ± 22.78 $^{ab}$ | **24.7 ± 10.98 $^b$** | 37.6 ± 21.14 $^{ab}$ | 33.0 ± 19.54 $^{ab}$ | 31.7 ± 19.84 $^{ab}$ | 37.5 ± 22.04 $^{ab}$ | 49.5 ± 20.93 $^{ab}$ | 57.9 ± 25.31 $^a$ | | 25–150 |
| TAN (mg/L) | **0.1 ± 0.16 $^b$** | **0.1 ± 0.09 $^b$** | **0.1 ± 0.09 $^b$** | **0.2 ± 0.18 $^{ab}$** | **0.2 ± 0.14 $^{ab}$** | **0.2 ± 0.13 $^{ab}$** | **0.2 ± 0.19 $^{ab}$** | **0.2 ± 0.12 $^{ab}$** | **0.3 ± 0.24 $^{ab}$** | | <0.01 |
| BOD (mg/L) | **4.7 ± 1.09 $^a$** | **4.6 ± 0.80 $^a$** | **4.4 ± 1.04 $^{ab}$** | **4.1 ± 1.18 $^{abc}$** | **3.5 ± 1.07 $^{abcde}$** | **4.0 ± 0.75 $^{abcd}$** | **3.7 ± 0.81 $^{abcde}$** | **3.9 ± 0.93 $^{abcde}$** | **3.3 ± 0.54 $^{bcde}$** | | 1–2 |
| NO$_3^-$ (mg/L) | 0.2 ± 0.07 $^a$ | 0.3 ± 0.06 $^a$ | 0.3 ± 0.07 $^a$ | 0.3 ± 0.079 $^a$ | 0.2 ± 0.07 $^a$ | 0.2 ± 0.04 $^a$ | 0.2 ± 0.07 $^a$ | 0.2 ± 0.11 $^a$ | 0.2 ± 0.08 $^a$ | | 0.1–4.5 |
| PO$_4^{3-}$ (mg/L) | 0.2 ± 0.18 $^a$ | 0.2 ± 0.21 $^a$ | 0.2 ± 0.15 $^a$ | 0.2 ± 0.15 $^a$ | 0.2 ± 0.12 $^a$ | 0.2 ± 0.19 $^a$ | 0.3 ± 0.21 $^a$ | 0.2 ± 0.18 $^a$ | 0.3 ± 0.22 $^a$ | | 0.05–0.5 |
| TC (MPNx$10^3$/100 mL) | **28.6 ± 37.5 $^a$** | **19.3 ± 15.96 $^a$** | **25.6 ± 44 $^a$** | **23.2 ± 35.6 $^a$** | **24.9 ± 32.65 $^a$** | **19.58 ± 26.49 $^a$** | **19.14 ± 15.68 $^a$** | **26.7 ± 37.4 $^a$** | **19.66 ± 16.54 $^a$** | | 0.1–5 |

The different superscripts $^{a, b, c, d, e, f}$ illustrate the statistical difference in water quality parameters. The bold values indicate concentration levels that are beyond the desirable ranges as per the guide.

Temperature is a significant factor in the aquatic environment that regulates various biological and physicochemical processes. In this study, the temperature of the different sampling sites was more or less similar, ranging from 29.9 °C to 31.0 °C (Table 2), and this was within the desirable range for aquaculture between 20–32 °C [36]. The mean surface water temperature ranged from 28.64 ± 1.48 °C in An Giang during the dry season to 29.69 ± 0.91 °C in Can Tho province during the wet season, all of which are within the recommended ranges. The highest value of temperature was in AG7, and the lowest value was recorded in CT1. The ANOVA test showed that the temperature values were not of significant difference ($p > 0.05$). During this study, the temperature values were close with a range of 29.9 ± 1.94 °C to 31.0 ± 1.27 °C. The temperature values were higher than those carried out by Giao [26], who reported a range of 26.8 to 29.4 °C in the Hau River. Generally, the temperature of the surface area is influenced by factors like flow, altitude, latitude, depth of water bodies, air circulation, and tree cover [37]. Higher temperatures are known to increase mineralization and release nitrogen, phosphorus, and carbon from soil organic matter [38], and thus the close relationships between water temperature and the rates of physiological processes also affect the food intake required to meet the demands for chemical energy and substrates necessary for survival. This factor therefore significantly affects aquaculture production [39]. On the other hand, colder water temperatures may reduce nutrient digestibility by reducing digestion rates, increasing gut transit time, and lowering gastrointestinal evacuation rates [40]. Moreover, an increase in run-off and erosion due to higher intensity of precipitations increase pollutants transport, especially after a drought period [41,42].

It is vital to maintain the aquatic resource within a pH range of 6.5–8.5 because any variation may lead to the destruction of the aquatic organisms [27,43]. The highest pH value was 7.6, noted in AG5, while the lowest was in AG7 and AG8 at 7.0 (Table 2) in An Giang Province. The values of pH from all the sampling sites are within the desirable range for aquaculture, which is 6.5 to 9 [44], and from the results, An Giang had the lowest pH value of 7.1 ± 0.25, and this was during the wet season, whereas Can Tho province showed the highest pH of 7.43 ± 0.39 and this was during the dry season. This was in agreement with other reports in similar water bodies and was within the allowable range of QCVN 08-MT: 2015/BTNMT (6.5–8.5) [26]. ANOVA test showed that the pH values were not significantly different ($p > 0.05$) between the sampling sites. However, in the present study, the pH was higher than the previous research carried out in the Hau River by Giao [26], who reported that a range of 6.7 to 7.1. pH variation might result from the discharge of wastewater, photosynthesis (whereby the photosynthetic activities consume dissolved carbon dioxide), and other metabolic processes [45]. It may be attributed to the introduction of silt into the river by rainwater or the mixing of the fast-flowing water as it moves downstream [46]. Generally, the pH increases during the dry period and decreases during the rainy season, meaning that seasonal variations are a critical factor that affects pH concentration [47] and subsequently, if not managed, will negatively impact aquaculture production [48].

The Dissolved oxygen values varied from 3.7 to 5.7 mg/L (Table 2) in different sampling sites, whereby the highest value was recorded in AG3 and AG5. In contrast, the lowest value was recorded in Can Tho province as 3.79 ± 0.92 mg/L during the wet season, whereas the highest mean value was realized in An Giang with 4.94 ± 1.05 mg/L during the dry season. Some sampled sites had values below the desired range for aquaculture (5–15 mg/L) [49] and therefore had possibilities of causing high levels of hypoxia. The ANOVA test showed that there were significant differences ($p < 0.05$) between the sampling sites. Oxygen is one of the most important environmental parameters that directly affect production and growth through metabolism and environmental conditions.

Also, oxygen has excellent effects on nutrient availability, and solubility and its deficiency can harm the cultured organisms or even increase toxic metabolites [50]. The dynamic nature of Dissolved oxygen results from the interaction of three factors. First, oxygen is not very soluble in water, so water has only a limited capacity to "hold" oxygen.

Second, the rate of oxygen use by fish, plankton, and organisms living in the pond mud can be high. Third, oxygen diffuses very slowly from the atmosphere into calm water. The combination of these three factors—limited solubility, rapid use, and slow replenishment can cause rapid changes in Dissolved oxygen concentrations [51]. The behavior of dissolved oxygen in most aquaculture ponds is especially complicated because of the intense biological activity in ponds receiving high feed inputs [52].

The low values of the dissolved oxygen at sampling site CT9 can be attributed to fertilizer run-off from the rice fields, organic waste, and mainly the untreated sewage from the industrial activities. A decrease in oxygen levels resulted from the pressure and discharge of high amounts of organic matter in the water. The reduction in dissolved oxygen along River Hau indicates that the deoxygenation rate caused by the biological decomposition of organic matter is higher than the re-oxygenation rate from the atmosphere [53,54].

The alkalinity in this study varied from 60.7 to 67.5 mg/L (Table 2), with the highest levels in AG8 and the lowest levels in AG1. The overall mean values showed that An Giang had the lowest alkalinity levels of $49.92 \pm 13.39$ mg/L, and this was during the wet season, whereas Can Tho province had the highest alkalinity value of $64.04 \pm 8.10$ mg/L in the dry season. Nevertheless, these levels were within the desirable range of aquaculture, which is 25–100 mg/L [44]. Analysis using one-way ANOVA showed no significant differences in alkalinity ($p > 0.05$). Alkalinity measures the amount of water that neutralizes or buffers acids using carbonate, bicarbonate ions, and hydroxide. This helps in protecting the aquatic organisms from significant pH fluctuations. Free carbon dioxide forms enormous amounts of carbonic acid, which is weak when there is no buffering capacity, leading to a decrease in pH to about 4.5 during the night. The pH values can increase above 10.0 due to high levels of photosynthesis, whereby the phytoplankton consumes most of the free carbon dioxide [55]. Besides, the high concentration of sewage and industrial waste causes high alkalinity in the polluted water. A good example is a study carried out in River Nyando [56], focusing on the agricultural and industrial activities taking place around the river. They found out that the fertilizers used in agriculture impacted the river's water quality, leading to high alkalinity levels as the river was flowing downstream.

During the study, salinity levels varied between the sampling sites with a range of 113.6 to 153.6 mg/L (Table 2), which were within the acceptable ranges in aquaculture (0–5000 mg/L) [55]. The highest concentration was noted in AG10, while the lowest was recorded in CT8. ANOVA test showed that the salinity values were significantly different ($p < 0.05$) between various sampling sites. The salinity levels in this study were within the desirable ranges of aquaculture, although attention needs to be paid to prevent salt intrusion from the sea. According to a survey carried out in the Mekong Delta of Vietnam by Tuan [57], saline waters affect approximately 2.1 million hectares of the coastal region through salt intrusion during the dry season, which reaches more than 60 km inland. This is a warning for Delta protection to avoid further salt intrusion in future because the Mekong River Commission identified salinity as one of the major water quality issues in the Lower Mekong Basin [58]. Observations from other studies have shown that laboratory measures of acute salinity tolerance can reflect the maximum salinity that macroinvertebrates and fish species inhabit, which is a key factor to note in the production of freshwater fish species [59].

In this study, the values of TAN ranged between 0.1 and 0.4 mg/L (Table 2), with the highest value being recorded in AG7. The TAN levels were above the desirable ranges for aquaculture, which is significant ($p < 0.01$) [49]. ANOVA test showed that there were significant differences between the sampling sites. Can Tho province registered the lowest TAN value of $0.23 \pm 0.16$ mg/L in the wet season while An Giang had the highest TAN levels of $0.41 \pm 0.49$ mg/L. The TAN levels reported in this study were higher than those reported from a study in the Hau River by Lien [60] that reported a range of 0.01–1.45 mg/L. The increase can result from increased waste disposal from the surrounding industries and fish processing factories into the river over the years [6]. Therefore, these wastes should be treated before they are discharged into the Hau River to sustainably undertake aquaculture production.

The TSS values were between 24.7 and 57.9 mg/L (Table 2), with CT9 recording the highest values and CT3 recording the lowest values of TSS. However, results also show that Can Tho province had the lowest TSS levels of 21.25 ± 8.72 mg/L during the dry season, while the highest value was recorded in An Giang during the wet season with 45.25 ± 28.48 mg/L. The TSS levels in all the sampling sites were within the range of aquaculture, which is 25 to 150 mg/L [55]. Analysis using one-way ANOVA showed significant differences in TSS ($p < 0.05$). The TSS levels noted in this study are almost similar to the TSS levels reported in a recent survey by Giao [26] that describes them as high. This is because the levels are suitable for aquaculture, unlike drinking. The author attributes the high levels of TSS to run-off water from anthropogenic activities and agricultural activities such as rice farming and aquaculture. Additionally, erosion, stormwater run-off, and high content of alluvial during the rainy season could cause high TSS in the Hau River [28]. Mainly, these suspended materials originate from a combination of sewer overflow systems, erosion from landfills, roads, vehicles, erosion in stream channels, and from the construction sites [61]. In this study, TSS levels were low, thus suggesting that there was no heavy pollution or erosion that had taken place along the river.

The BOD levels were higher than the desired ranges for aquaculture (1–2 mg/L) [44] because they ranged from 2.7 to 4.7 mg/L (Table 2). The highest concentrations were noted in CT1, while the lowest levels were indicated in AG3 and AG8. The ANOVA test showed that there were highly significant differences in BOD between the sampling sites. The high BOD concentrations in this study can be attributed to high levels of organic contamination from the aquaculture farms, leaky sewer pipes and tanks, agricultural run-off, and non-point source pollution that support micro-bacteria growth [62]. During the study, the BOD levels were above the desired range for aquaculture, with higher concentrations recorded during the dry season. There is a severe increase in BOD concentrations during the dry season because the metabolism of both aerobic and anaerobic micro-organisms increases with an increase in temperature, and there is decreased water flow in the river. On the other hand, large amounts of freshwater are dilute the organic matter during the rainy season, thus leading to a decrease in BOD concentrations [53].

The range of nitrate levels was between 0.2 and 0.3 mg/L (Table 2), which were within the desirable ranges of aquaculture, which is 0.1 to 4.5 mg/L [44]. The results from ANOVA show that there was no significant difference between the different sampling sites. Nitrate-nitrogen is a very critical nitrogenous nutrient in fish culture [50]. In the current study, the nitrate levels are within the desirable ranges while the concentrations of nitrate were a bit higher than those reported in the previous studies in the Hau River in 2016 with approximately 0.1 mg/L [60] and the survey carried out by Giao [26] that ranged between 0.08 and 0.33 mg/L. During rainfall, the levels of nitrate rise due to run-off from various agricultural activities that carry fertilizers to the water bodies, thus leading to pollution. The increase in nitrate causes eutrophication leading to low oxygen levels that affect the aquatic organisms since they cannot tolerate anaerobic conditions. High nitrate levels recorded in surface waters originate from human activities and differ with land use. High nitrate concentrations observed in many river systems may be due to diffuse urban and agricultural run-off sources and point discharge from sewage treatment plants. The sources of nitrate in the surface water are mainly the dry and wet deposition of nitric acid formed in the atmosphere through the nitrogen cycle. Human activities are the most significant contributors to the high levels of nitrate in the surface waters though they differ depending on the type of land use [63].

The phosphate levels ranged between 0.1 and 0.3 mg/L (Table 2), which was within the desirable ranges of aquaculture, which is 0.05 to 0.5 mg/L [44]. There was no significant difference among the sampling sites. Domestic wastewater, mainly those containing detergents, industrial effluents, and fertilizer run-off, contributes to elevated phosphates levels in surface waters. High concentrations of phosphates can indicate the presence of pollutants that are primarily responsible for atrophic (people's activities) conditions [64]. In the present study, the levels of phosphate were still in the suitable concentrations desired

for aquaculture. However, they were relatively higher than the values reported in the previous years, which were approximately 0.1 mg/L and a range of 0.04 to 0.10 mg/L [46], respectively [26,60].

The total coliform values ranged between 14.2 and 31.5 MPN $\times 10^3$/100 mL (Table 2), which means they were higher than the desirable range for aquaculture [55]. There was no significant difference among the sampling sites from the ANOVA results. During this study, the levels of total coliform were way above the desired levels for aquaculture. The high concentrations of total coliform noted during this study were consistent with those reported [26], who pointed out that An Giang province had higher values than Hau Giang province, whereby An Giang values ranged from 31,835 MPN/100 mL to 86,338 MPN/100 mL. These high levels of total coliform in the Hau River can be attributed to the release of high amounts of organic waste into the river, especially from the aquaculture farms, rice fields, fish processing industries, and other industrial activities. Similarly, it was [65] who pointed out that total coliforms are affected mainly by organic matter and are primarily affected by phosphate, phosphorus, and suspended solids when carrying out a study in the Nakdong River [65].

### 3.2. The Accumulation Factor and River Recovery Capacity (RRC) of the Water Quality Parameters

The accumulation factor and River Recovery Capacity (RRC) of the water quality parameters analyzed are presented in Table 3. The accumulation factor of the parameters revealed that BOD and $PO_4^{3-}$ of the downstream were 1.53 and 1.52, respectively, more than the upstream. TAN showed the highest percentage recovery of about 54.79%, while pH showed the lowest recovery of about 1.55% in water downstream. In this study, the accumulation factor of the parameters showed higher values downstream than upstream, thus indicating the anthropogenic impacts on the river. The River Recovery values for the water quality parameters showed a change in the values at the upstream sites and those at the tail site. Most of the parameters had low recovery values, which indicates that these pollutants are being released into the river in large quantities, exceeding the removal carrying capacity of the Hau River [66]. In other words, the nutrients loaded are in levels higher than the river recovery capacity.

**Table 3.** Accumulation factor and river recovery capacity for water quality parameters of the Hau River.

| Parameter | Temp | pH | Alk | Dissolved Oxygen | Sal | TSS | TAN | BOD | NO$_3^-$ | PO$_4^{3-}$ | TC |
|---|---|---|---|---|---|---|---|---|---|---|---|
| AF | 0.98 | 1.02 | 1.05 | 0.84 | 0.91 | 1.29 | 0.65 | 1.53 | 1.21 | 1.52 | 1.23 |
| RRC (%) | −2.36 | **1.55** | 4.39 | −18.76 | −10.39 | 22.69 | **−54.79** | 34.71 | 16.87 | 34.18 | 18.54 |

AF: Accumulation Factor; RRC: River Recovery Capacity. Bold values indicate the highest and lowest percentages of recovery.

### 3.3. Principal Component Analysis (PCA)

The principal component analysis was used to obtain the essential water quality parameters and factors affecting the water quality of the Hau River. In this study, there were some difficulties in extracting definite conclusions due to complex relationships. However, the principal component analysis helped retrieve the information to some extent and explain the structure of the data in detail based on temporal characteristics, not forgetting to elucidate the relationship between different variables. Therefore, PC is applicable in identifying the number of sources having effects on environmental monitoring [67]. The correlation between the principal components and water quality parameters (the initial data variables) is expressed by loading or weighing factors [67]. The absolute values of the weighing factors differ in meaning because when it is greater than 0.75, it means that there is a close correlation between the water quality indicators and the main component. When the absolute value of the weighting factor is between 0.75 and 0.5, there is an average correlation, while 0.5 to 0.3 means there is a weak correlation [68].

MINITAB version 19.1 was used to analyze the principal component to determine the leading principal components from the original variables. The mean value of each of

the 11 water quality parameters was used in the principal component analysis. Table 4 illustrates the results of the analysis. Eleven factors contributed to the overall interpretation of the change in the Hau River's surface water quality from An Giang to Can Tho. PC1 had the most considerable contribution, which was 17%, while the least was PC11 with a 3.5% contribution. In research, it is only the principal components with eigenvalues greater than one that are considered significant. In this study, based on the scree plot, labeled Figure 2, the 11 parameters were reduced to 5 factors from the scree plot's leveling off points. The eigenvalues of PC6 to PC11 were less than one, which could be ignored because they were less than unity. However, PC1, PC2, PC3, PC4, and PC5 had eigenvalues greater than one; therefore, be helpful. PC1 had the largest eigenvalue of 1.87, accounting for approximately 17% of the total variance, followed by PC2, PC3, PC4, and PC5, whose eigenvalues are 1.66, 1.49, 1.16, and 1.02 with a total variance of 15.1%, 13.6%, 10.5%, and 9.3%, respectively.

**Table 4.** Principal component analysis for water quality on the Hau River in 2019.

| Parameter | PC1 | PC2 | PC3 | PC4 | PC5 | PC6 | PC7 | PC8 | PC9 | PC10 | PC11 |
|---|---|---|---|---|---|---|---|---|---|---|---|
| Temp | 0.395 | 0.081 | 0.29 | −0.266 | 0.309 | −0.323 | **0.51** | −0.158 | −0.078 | 0.288 | 0.331 |
| pH | 0.276 | **−0.42** | −0.312 | 0.023 | −0.211 | 0.236 | 0.315 | 0.177 | **−0.462** | 0.333 | −0.308 |
| Alk | **0.419** | **0.441** | −0.209 | −0.108 | 0.044 | −0.023 | 0.031 | −0.385 | 0.09 | −0.201 | **−0.612** |
| DO | **0.516** | −0.235 | −0.072 | 0.108 | −0.138 | 0.303 | −0.272 | −0.392 | −0.072 | −0.288 | **0.484** |
| Sal | 0.288 | 0.232 | **0.466** | −0.078 | 0.018 | −0.028 | −0.375 | **0.45** | **−0.509** | −0.16 | −0.103 |
| TSS | −0.275 | −0.066 | 0.215 | **−0.559** | −0.389 | −0.02 | −0.251 | **−0.471** | −0.217 | 0.27 | −0.074 |
| TAN | −0.294 | 0.243 | 0.241 | **0.454** | −0.214 | 0.076 | **0.434** | −0.307 | **−0.431** | −0.262 | 0.031 |
| BOD | −0.171 | −0.026 | **−0.557** | −0.129 | 0.239 | **−0.504** | −0.128 | −0.059 | **−0.473** | −0.253 | 0.158 |
| NO$_3^-$ | 0.127 | 0.297 | −0.21 | −0.223 | **−0.707** | −0.183 | 0.221 | 0.323 | 0.156 | −0.162 | 0.253 |
| PO$_4^{3-}$ | −0.187 | 0.288 | −0.173 | **−0.458** | 0.29 | **0.668** | 0.184 | 0.116 | −0.117 | −0.135 | 0.179 |
| TC | −0.035 | **−0.528** | 0.253 | −0.322 | 0.02 | −0.08 | 0.273 | 0.05 | 0.114 | **−0.638** | −0.221 |
| Eigenvalue | 1.87 | 1.66 | 1.49 | 1.16 | 1.02 | 0.8 | 0.78 | 0.66 | 0.63 | 0.56 | 0.38 |
| Variation (%) | 17 | 15.1 | 13.6 | 10.5 | 9.3 | 7.3 | 7.1 | 6 | 5.7 | 5.1 | 3.5 |
| Cumulative variation (%) | 17 | 32 | 45.6 | 56.1 | 65.4 | 72.7 | 79.8 | 85.7 | 91.4 | 96.5 | 100 |

The bold values indicate a weak correlation between the parameters and the principal components

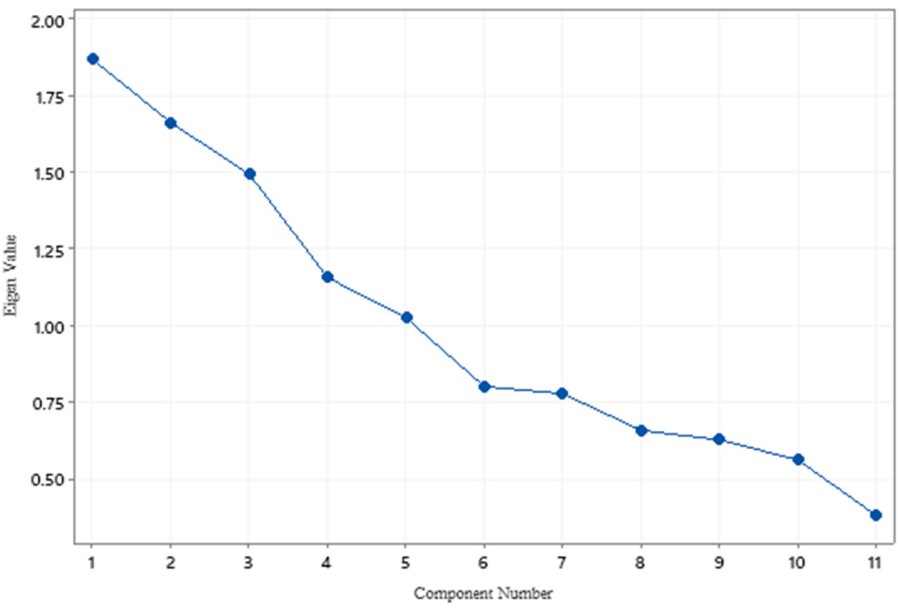

**Figure 2.** Scree plot of eigenvalues in the Hau River.

Alkalinity (positive) was weakly correlated to PC1, while Dissolved oxygen (positive) was moderately correlated to PC1. PC2 was weakly contributed by both pH (negative) and alkalinity (positive), while total coliform (negative) was moderately correlated to it. PC3 was weakly and moderately contributed by salinity (positive) and BOD (negative), respectively. For PC4, TAN and PO$_4^{3-}$ were weakly correlated, while TSS (negative)

was moderately correlated. On the other hand, PC5 was moderately contributed by nitrate (negative). Nevertheless, PC6–PC11 was retained for discussion because they were correlated with the parameters. PC 6 was moderately correlated with both BOD (negative) and $PO_4^{3-}$ (positive), respectively, PC7 was weakly correlated with TAN (positive), PC8 was weakly correlated with TSS (negative), PC9 was weakly correlated with pH (negative), TAN (negative), BOD (negative) and moderately correlated with Salinity (negative). On the other hand, PC10 was moderately correlated with total coliform (negative), while PC11 was moderately correlated with Alkalinity (negative) and weakly correlated with Dissolved oxygen (positive). From the above values, it can be concluded that the Hau River's surface water quality deterioration was due to five major sources (PC1, PC2, PC3, PC4, and PC5) and six minor sources (PC6 PC7, PC8, PC9, PC10, and PC11).

At PC1, hydrological factors (alkalinity and Dissolved oxygen) were the sources of water quality deterioration. PC2 had similar origins, although it was also affected by artificial sources such as human activities from residential areas through the release of sewerage wastewater. PC3 potentially represents the source of impact related to organic matter, resulting from anthropogenic activities and other wastewater sources [69]. PC4 represents artificial sources such as uneaten feeds and wastewater from aquaculture facilities, industrial activities, and other sources such as agricultural run-off from rice farms. At PC5, agricultural activities were the source of pollution on the Hau River.

In summary, the fluctuation of surface water quality in the Hau River results from various sources, including anthropogenic activities, agricultural run-off, industrial activities, aquaculture activities (Domestic and urban waste generation), and hydrological factors. These sources are in line with the previous studies that have been carried out in Mekong Delta because it was concluded that the sources affecting water quality were aquaculture, livestock, agriculture production, overflow rainwater, urban and residential areas, hydrological factors, tourism, and industrial activities [28]. There is a need for more surveys on the Hau River to provide proper measures to ensure the elimination of contaminants in the surface water. Other parameters should be collected for PCA analysis before making a final decision in the monitoring task of the Hau River because this study covered only 11 parameters, therefore, giving a partial representation of water quality parameters that may influence the Hau River's water quality.

### 3.4. Cluster Analysis

The cluster analysis is a multivariate statistical technique, which assembles objects based on their similarity [70]. It plays a crucial role in providing a solution to the classification problems by placing different variables into groups so that members of the same cluster have a substantial degree of the association while the members of different clusters have a weak degree of association [71]. In this study, the cluster analysis used averaged values of water quality parameters at 19 various sampling sites in the Hau River in both An Giang and Can Tho provinces. The results of the grouping are shown in Figure 3. There was a strong association between the spatial and temporal variations of principal pollution factors, as demonstrated by the cluster analysis. This further indicated that human activities have significant effects on the water quality of the Hau River since they vary both spatially and temporally. The dendrogram shows the pollution status at the sampling sites, and it gives a summary of the clustering processes and their proximity. Cluster analysis (CA) was used to identify any similarity between the nineteen sampling sites by generating a dendrogram, then followed by grouping the sampling sites according to the percentage of their similarity and difference concerning the water quality parameters. The analysis of the similarity level of study sites from 4.349 to 100% was carried out to indicate the intensity of the relationship between sites as a cluster. It can be noticed that the sampling sites could be separated into six groups at a similarity level of 68.12%, which include Group 1(AG1), Group II (AG2-AG6, AG10), Group III (AG7-AG9), Group IV (CT1, CT6, and CT7), Group V (CT2-CT4) and Group VI (CT5, CT8, and CT9)

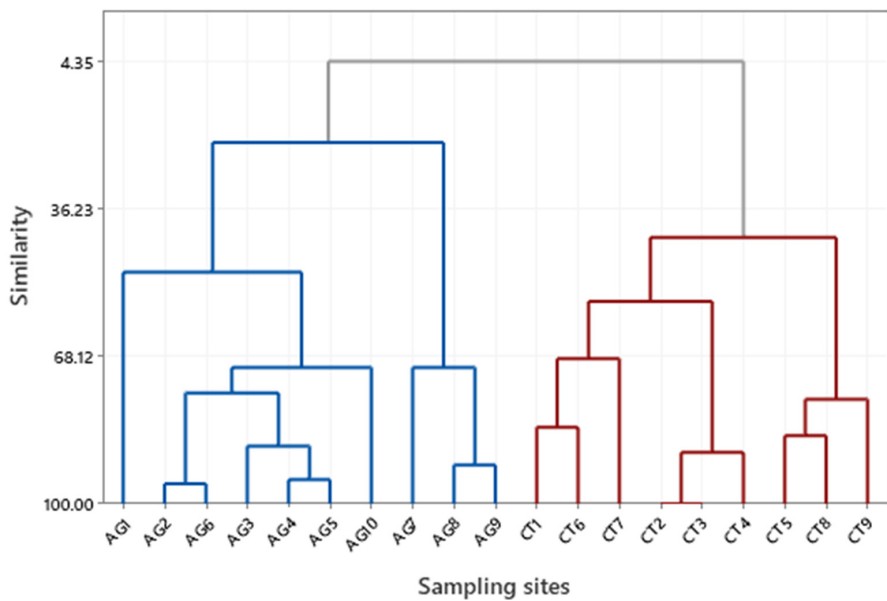

**Figure 3.** Clustering monitoring sites in the Hau River in 2019.

This separation is due to a higher concentration of TAN, BOD, and coliforms across all the sampling sites (Table 1). There could be a significant source of pollutants affecting the water quality, such as commercial fish ponds and waste from residential areas. From the above clustering, it can be deduced that the sampling sites based on their level of similarity can be reduced from 19 sampling sites to 12 sites and still get meaningful data that depict a true reflection of the water quality parameters of the Hau River in the two provinces which can thus reduce on the costs spent in collecting data from the nineteen (19) sites. Based on the grouping of water quality presented in Figure 4, the number of water quality parameters could be reduced from 11 to 7, including Temp/Alk/Sal, pH/Dissolved oxygen, TSS/TC, TAN, BOD, $NO_3^-$, and $PO_4^{3-}$ from the Hau River. However, other water quality parameters need to be evaluated to plan and manage water quality for sustainable aquaculture production.

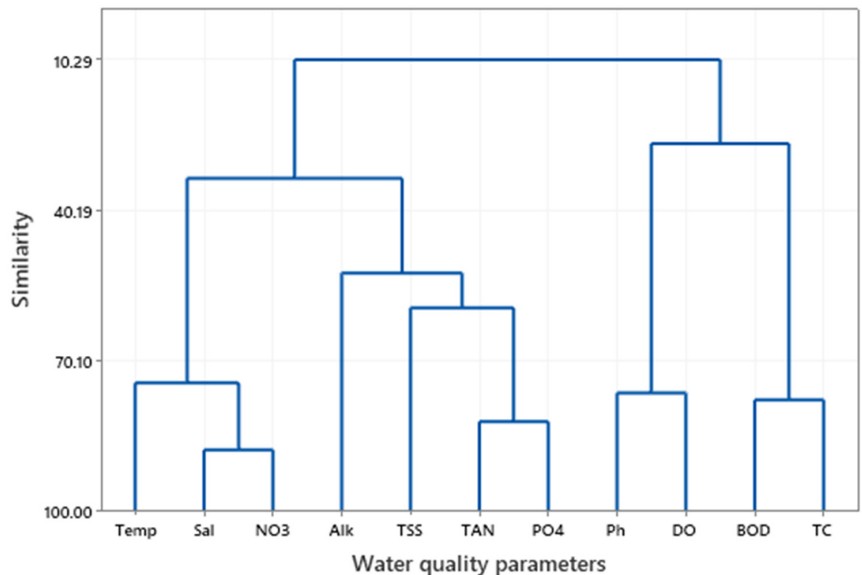

**Figure 4.** Clustering water quality parameters.

## 4. Conclusions

This study, along with the Hau River, demonstrated that the river has high BOD and total coliform levels while Dissolved oxygen levels are slightly below the desirable ranges for aquaculture. The low Dissolved oxygen values that varied from 3.7 to 5.7 mg/L in different sampling sites, e.g., at CT9, which was $3.79 \pm 0.92$ mg/L, are as a result of high BOD reported being from 2.7 to 4.7 mg/L higher than the desired ranges for aquaculture (1–2 mg/L). This was more so in areas surrounded by Pangasius fish farms and agricultural runoffs, which call for wastewater treatment provisions to avoid degrading the water quality in the Hau river for its sustainable utilization for aquaculture production. According to the results from PCA, water quality parameters that included: temperature, Dissolved oxygen, pH, Alkalinity, Salinity, TAN, TSS, BOD, $NO_3^-$, $PO_4^{3-}$ and total coliform is essential in indicating the status of the Hau River's water quality because they affected the surface water quality at various sampling sites. Wastes from aquaculture facilities, agricultural areas, and industries affected the water quality of the Hau River, which is explained by PC1, PC2, PC3, PC4, and PC5. The cluster analysis makes it possible to reduce the sampling sites from 19 to 6, which cuts the costs and obtains the data required to draw a reliable conclusion. The treatment of wastes before disposal is vital in maintaining the water quality for sustainable aquaculture production. Therefore, multivariate statistical techniques are crucial in assessment, evaluation, and monitoring to maintain the quality of water resources.

**Author Contributions:** Conceptualization, U.N.V.; Data curation, U.N.V. and T.G.H.; Formal analysis, F.G.M. and H.V.A.; Funding acquisition, U.N.V.; Investigation, F.G.M. and H.V.A.; Methodology, U.N.V.; Resources, U.N.V. and T.G.H.; Supervision, T.G.H.; Visualization, U.N.V. and H.K.N.; Writing—original draft, F.G.M. All authors have read and agreed to the published version of the manuscript.

**Funding:** This study was funded in part by the Can Tho University Improvement Project VN14-P6, supported by a Japanese Official Development Assistance (ODA) loan. The APC was funded by this Project in this section.

**Data Availability Statement:** The data that support the findings of this study are openly available in LRC Digital Repository at https://dspace.ctu.edu.vn/jspui/handle/123456789/38886 (18 September 2020).

**Acknowledgments:** The authors would like to thank the undergraduate students of Batch 42 studying at College of Aquaculture and Fisheries, Can Tho University for their help in field sampling.

**Conflicts of Interest:** The authors declare no conflict of interest.

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
