# Peer review of "Assessment of Water Quality for Aquaculture in Hau River, Mekong Delta, Vietnam Using Multivariate Statistical Analysis"

_water, doi:10.3390/w13223307_

Round 1
Reviewer 1 Report
Comments and Suggestions for Authors in attachment.

Author Response
Thank you Reviwer for accepting the revised manuscript withou further correction or revision.
Reviewer 2 Report
The manuscript has undergone revisions by the authors based on the reviewer’s initial comments.
The authors have addressed all of the reviewer comments and provided adequate feedback/commentary regarding changes made to the revised manuscript. Changes made represent a significant improvement to the manuscript, yet additional formatting and presentation changes based on the revised manuscript are necessary for publication.
General Comments:
Ensure consistent spelling through manuscript, e.g. spatiotemporal or spatio-temporal.
Ensure consistent spacing between values and units, e.g. see (1) Lines 95–99, (2) 208 and 213, and (3) Line 362,363, 370, etc…
Ensure consistent formatting for all cited works in reference listing. For example, publication year are presented in three varying formats, e.g. (year), 2008, and year, etc.
Ensure use of italics where necessary throughout reference listings, e.g. Page 18, Line 665 (Seriola lalandi).
Specific Comments:
Page 2, Line 93: Suggest amending “billion meter cubed” to “billion m3”
Page 3, Figure 1: (1) Suggest selection of different colouration of global scale inset and (2) Reduce size or inset to ensure AG1 is visible in figure, currently reads “G1” not “AG1”.
Page 3, Line 108: Ensure Table 1 caption and table are presented together.
Page 4, table: Ensure table is presented with a table caption. Additionally, suggest presenting table columns wide enough to ensure all cell entities are presented in single row, with exception of the description column.
Page 5, Line 156: (AF) not necessary, already presented in previous text, i.e. Line 155.
Page 7, Line 242: Correct [Error! Reference source not found.], amend insert correct cross-reference.
Page 8, Line 307: Ensure correct tense here and throughout manuscript, e.g. amend “…The province register the lowest…” to “…The province registered the lowest…”
Page 10, Line 384: Insert space between paragraph text and table caption.
Page 10, Line 385-386: Ensure Table 2 caption and table are presented together.
Page 10, Lines 390-391: Ensure table footnotes are presented underneath, in association with, the table.
Page 10, table: Ensure table is presented with a table caption. Additionally, suggest presenting table in ‘landscape’ orientation for improved presentation, i.e. by providing wider rows will ensure all cell entities are presented in single row.
Page 12, Line 433: Amend text “,…la-bled figure 2…” to “,…la-bled Figure 2…” in addition, look to present Figure 2 closer to the text cross-referencing Figure 2, i.e. closer than page 15. Present figures and cross-referencing text closer where possible.
Page 12, Line 440: Ensure consistent use of acronyms, e.g. dissolved oxygen or DO, here and throughout manuscript (e.g. Line 549).
Page 13, Ensure Table 4 caption and table are presented together.
Page 14, Line 504: Ensure table footnotes are presented underneath, in association with, the table.
Author Response
Thank you very much Reviewer 2 for your detail comments on formating. We have corrected all these suggestions and also checked English. The corrections are ressponded in the attached file.

This manuscript is a resubmission of an earlier submission. The following is a list of the peer review reports and author responses from that submission.
Round 1
Reviewer 1 Report
WATER-1250073
Review of Assessment of water quality for aquaculture in Hau River, Mekong Delta, Vietnam using multivariate statistical analysis
Manuscript type: Article
The manuscript describes and discusses the study analyzed the water quality of the Hau River using standard procedures where 11 water quality parameters were analysed using Principal component analysis (PCA) and Cluster Analysis (CA). River water quality was sampled at 19 sampling sites every month in the year 2019. The findings showed high levels of biological oxygen demand (BOD), total suspended solids (TSS) and total coliforms (TC), all year round. The dendrogram of similarity between sampling sites showed a maximum similarity of 95.6%. The trend in accumulation factor showed that the concentrations of TSS, BOD, and PO43- in the downstream were 1.29, 1.53, and 1.52, times, respectively, greater than the upstream levels.
This study is an interesting study relevant for publication and utilizes appropriate study design and methodologies. However, a number of items must be firstly addressed before publication. The authors are encouraged to revise the manuscript and resubmit.
REJECT - Reconsider after major modifications.
Below are general and specific points requiring attention.
- Moderate English changes required
- The manuscript contains relevant sections as necessary however the authors need to ensure correct formatting throughout the manuscript, including, headers (including specific title page header), footers, and line numbering are utilized.
- The results section is presented in a straightforward manner, however the discussion of results requires improved and additional commentary.
- The authors should look to refer to, and cite, where applicable specific national and regional water quality standards/criteria that the results findings can be compared to, in addition to the aquaculture-use specific references used throughout the manuscript.
- It is suggested that given rainfall is a very important contributing parameter in river water quality, which the authors acknowledge, this should be presented with greater emphasize. Suggested additional results and discussion includes statistical analysis/comparison of datasets in dry season vs wet season and time-series of selected water quality parameters which illustrate increased values/concentrations during the wet vs dry seasons. Months of the wet and dry seasons should also be reported.
- Discussion section reads as results section. There is a need to put the results into context with commentary of how the nominated factors influencing the water quality parameters specifically contribute to the measured conditions. Moreover, discussion of the relative importance/contribution of factors when discussed with associated water parameters is encouraged with specific associated commentary.
- Are the authors able to provide any information that aquaculture operations do/should undertake to limit the influence of these practices on water quality that in turn would benefit water quality for aquaculture uses in both the introduction and discussion sections. Given the focus of manuscript is on aquaculture, specific information is deemed important. Is it possible to inform readers of sites which are closer to aquaculture sites (and what type/species) and thus the water quality in the river maybe more impaired at such sites due to discharges? Lastly, are optimal water quality parameters cited and discussed specific to particular aquaculture practices/species?
Abstract:
Page 1, Second Sentence: 1) Introduce acronyms before use for BOD , TSS; 2) no need for capitals when including Total Coliform, amend to total coliforms. Change to total coliforms or TC, throughout mansucript.
Page 1, Forth Sentence: 1) Introduce acronyms before use for AF; amend text to include /values after concentrations, i.e. “…showed that the concentrations/values of TSS…”
- Introduction:
Page 2: Introduce acronyms before use, e.g. GDP (Gross Domestic Product), PCA (principal component analysis) and CA (cluster analysis)
Page 2, Aims Section: In addition to existing aim text, include the study also places emphasis on assessment of water quality for aquaculture practices as per the manuscript title.
- Methods
Page 2, 2.1 Study Location: Suggest cross reference Fig 1 with introduction of the study region.
Page 3, 2.1 Study Location: Use consistent units and presentation of units when discussing flow
Page 3, 2.1 Study Location: Last Sentence: Confirm/correct number of sampling sites. Understood to be 19 NOT 12.
Page 4, Table 1: Provide indicators of which sites are upstream sites and which sites are downstream as references within the results and discussion text.
Page 5, Figure 1: 1) Add insets providing international and regional context to better illustrate the study region, 2) Add place names, 3) improve image resolution to provide a clearer image for ease of viewing.
Page 5, 2.2 Sampling and Analysis: Here and where else necessary spell out acronyms before use. For example, DO (dissolved oxygen), TSS (total suspended solids, BOD (biological oxygen demand), TAN (total ammonia nitrogen), etc… Thereafter, where appropriate be consistent with use of acronyms.
Page 5, 2.2 Sampling and Analysis: Please indicate rainfall observations in the area for the acknowledged rainy season and also dry season values from any weather stations/gauges.
Page 6, 2.2 Sampling and Analysis: Can the authors look to provide any quality assurance/quality control information regarding the analytical techniques. For example, method detection limits, use of blanks, standards, error, etc.,
Page 6, 2.3 Statistical Analysis: Suggest reiterating the upstream and downstream sample sites
- Results
Ensure consistency with p not P when citing statistical results, e.g. p<0.05 or P<0.05
Ensure consistency with spaces between units, symbols and text, e.g. p>0.05 or p. 0.5; 2.7 to 4.7 mg/L or .1 to 4.5mg/L, etc…
- Discussion
Page 9, 4.1 Temperature: Please look to quantify temperature values described as moderate
Page 9, 4.2 pH: Heading to read ph NOT PH
Page 9, 4.2 pH: First sentence to be removed
Page 9, 4.4 Alkalinity and 4.5 Salinity: Look to provide more specific discussion regarding conditions and influencing factors within the study period/location
Page 9, 4.6 TAN: Ensure correct use of referencing formatting. Amend 52 to [52]. Correct here and elsewhere where necessary.
Page 9, 4.11 Total Coliforms: The first sentence is confusing. Please look to rewrite.
Page 12, Table 2: 1) Consider indication of upstream and downstream sample sites in a table row; 2) within the desirable column in table include reference [#]; 3) confirm TC units MPNx103/100mL to MPNx103/100mL; 4) indicate significance of bold parameter values
Page 13, upper paragraph: This paragraph appears separate from Section 4.11. Confirm this section does or does not require a heading.
Page 13, Table 3: indicate significance of bold parameter values
Page 13, 4.12 Principal Component Analysis (PCA): Look to move the sentences 4-9 to methods section.
Page 14, 4.12 Principal Component Analysis (PCA): Provide explanation of what is meant by ‘water quality complications’?
Page 14, 4.12 Principal Component Analysis (PCA): In addition to the 11 parameters measured, which are stated as being a partial representation of water quality parameters, can the authors provide information regarding what additional water quality parameters should be targeted as high priority in subsequent studies.
Page 14, Table 4: indicate significance of bold parameter values
Page 16, 4.13 Cluster Analysis: Ensure consistent use of italics in heading
Page 16, 4.13 Cluster Analysis: Please confirm the distinct separation of sites AG and CT, are these downstream and upstream?
Reviewer 2 Report
Comments and Suggestions for Authors in attachment.

Reviewer 3 Report
The manuscript water-1250073 entitled “Assessment of water quality for aquaculture in Hau River, Mekong Delta, Vietnam using multivariate statistical analysis” aimed to analyse the water quality of the Hau River using standard procedures and analysing data with Principal component analysis (PCA) and Cluster Analysis (CA).
The MS provides data of local interest without a substantial hypothesis. Also, the use of PCA or CA is not novel in the field of water quality assessment. I am sorry, and I know this is disappointing for the Authors, but I have to say that this work is not suitable for a high-quality journal as Water.